CAM-YOLO: tomato detection and classification based on improved YOLOv5 using combining attention mechanism

Appe Seetharam Nagesh nageshf25@gmail.com 1 2
G Arulselvi 1
GN Balaji 3
1 Department of Computer Science and Engineering, Annamalai University , Chidambaram , Tamilnadu , India
2 Department of Information Technology, CVR College of Engineering , Hyderabad , India
3 School of Computer Science and Engineering, Vellore Institute of Technology , Vellore , Tamilnadu , India
Moparthi Nageswara Rao
Electronic publication date: 2023 Jul 20
Publication date: 2023
Volume: 9
Electronic Location ID: e1463
Received 2023 Mar 7; Accepted 2023 Jun 6
Copyright: ©2023 Appe et al.
Copyright year: 2023
Copyright holder: Appe et al.
License: This is an open access article distributed under the terms of the Creative Commons Attribution License, which permits unrestricted use, distribution, reproduction and adaptation in any medium and for any purpose provided that it is properly attributed. For attribution, the original author(s), title, publication source (PeerJ Computer Science) and either DOI or URL of the article must be cited.
License URL: https://creativecommons.org/licenses/by/4.0/

Keywords: Object detection, Non-max suppression, Distance intersection over union, CBAM

Funding: The authors received no funding for this work.

==============================
Background

One of the key elements in maintaining the consistent marketing of tomato fruit is tomato quality. Since ripeness is the most important factor for tomato quality in the viewpoint of consumers, determining the stages of tomato ripeness is a fundamental industrial concern with regard to tomato production to obtain a high quality product. Since tomatoes are one of the most important crops in the world, automatic ripeness evaluation of tomatoes is a significant study topic as it may prove beneficial in ensuring an optimal production of high-quality product, increasing profitability. This article explores and categorises the various maturity/ripeness phases to propose an automated multi-class classification approach for tomato ripeness testing and evaluation.

Methods

Object detection is the critical component in a wide variety of computer vision problems and applications such as manufacturing, agriculture, medicine, and autonomous driving. Due to the tomato fruits’ complex identification background, texture disruption, and partial occlusion, the classic deep learning object detection approach (YOLO) has a poor rate of success in detecting tomato fruits. To figure out these issues, this article proposes an improved YOLOv5 tomato detection algorithm. The proposed algorithm CAM-YOLO uses YOLOv5 for feature extraction, target identification and Convolutional Block Attention Module (CBAM). The CBAM is added to the CAM-YOLO to focus the model on improving accuracy. Finally, non-maximum suppression and distance intersection over union (DIoU) are applied to enhance the identification of overlapping objects in the image.

Results

Several images from the dataset were chosen for testing to assess the model’s performance, and the detection performance of the CAM-YOLO and standard YOLOv5 models under various conditions was compared. The experimental results affirms that CAM-YOLO algorithm is efficient in detecting the overlapped and small tomatoes with an average precision of 88.1%.

Introduction

Tomatoes are a commercially significant horticulture product and harvesting, like many other crops, is a labour-intensive task. In past few years, there has been a great deal of interest in automating agricultural activities such as harvesting (Edan, 2014), spraying (Oberti et al., 2015) and pruning (Paulin et al., 2015). This has resulted in development of computer vision and image analysis systems to detect the vegetables and fruits.

The introduction of artificial intelligence in the agriculture field has drawn significant widespread interest, primarily in the design of the harvesting robots. It was designed to replace hand fruit picking, which is expensive, labour-intensive, and prone to human error. Nevertheless, the important phase of harvesting robots is automatic detection of fruits or other agricultural goods The detection accuracy data are generally used to direct a robot in picking the fruits However, developing an object recognition and detection system for fruit and vegetable detection that is as smart as humans is incredibly challenging due to a variety of factors such as non-uniform illumination, occlusion, and so on.

With the advent of deep learning, the velocity of object detection has dramatically accelerated in recent years (Jiao et al., 2019) and made one accomplishment after another during this period of rising and growing. In contrast to standard pattern recognition methods, the deep learning-based fruit detection approach can directly take the original image as input and the end-to-end structure is utilised to speed up the detection of the object rather than time-consuming steps like image pre-processing, feature extraction and classification. In addition to saving time and effort, the deep learning-based object detection technique also enables real-time judgement when detecting fruits.

Object detection method used to identify and locate objects within an image or video by using bounding boxes. Deep learning-based object detection methods uses convolutional neural networks (CNN) to carry out end-to-end, unsupervised object recognition and detection which eliminates the requirement for feature definition and separate feature extraction. Typically, object detectors using deep learning extract features from the input image. Object detection systems are categorised into two types: single-stage object detectors and two stage object detectors (Nguyen et al., 2020) shown in Fig. 1. The single-stage detectors (YOLO, SSD Zhang et al., 2018; Lin et al., 2020) provides high inference rates whereas two-stage detectors provide higher localization and identification accuracy (Mask RCNN, Faster RCNN, Fast RCNN, RCNN) Ren et al., 2017; Ma, Zhang & Xu, 2019).

Figure 1 Types of object detection systems.

The major contribution of this article is an attention mechanism based YOLOv5 model classifier algorithm for detecting and classifying the tomato fruits. The key points considered are:

(a) To create a dataset of 2,034 images which consists of both ripe and unripe tomatoes.

(b) To label the images manually for training propose.

(c) To modify the existing YOLov5 algorithm for tomato ripeness detection and classification.

Further, this article is organised as follows: ‘Introduction’ described about the introduction, related works are illustrated in ‘Related Work’, the proposed architecture in the ‘Methodology and Materials’ followed by the experimental results in ‘Experiments and Results’ and finally ‘Conclusions’ concludes the article.

Related Work

Deep learning algorithms, in particular, have been applied and tested more and more in recent years (Xu et al., 2020; Liu et al., 2019) for fruit detection. Machine learning provides an improved solution to issues like occlusion and green tomato recognition compared to traditional approaches since it is more reliable and accurate. Due to the difficulties of segmenting green tomatoes and distinguishing them from the background because of their similar color, the topic of green tomato detection is rarely explored.

Lü et al. (2014) deployed a support vector machine (SVM) trained exclusively on the RGB colour space for the purpose of identifying fruit and branches in natural sceneries. They claimed that their method outperformed earlier threshold-based techniques, achieving an accuracy of 92.4% for fruits. However, the results were susceptible to lighting effects.

Rahnemoonfar & Sheppard (2017) applied the algorithm for counting of fruits which is based on Resnet and Inception. The algorithm achieved overall prediction performance of 91% in real time. The algorithm may be able to count the fruits even if some of them overlap with one another also under shadow. However, the algorithm used for only counting fruits, but not implemented for detection.

Now, YOLO models are widely used in agriculture for applications like disease detection (Wang et al., 2022) fruit detection (Dai & Fan, 2022), pest detection (Liu & Wang, 2020; Wang, 2022), and grass identification (Saleem et al., 2022). Based on the above literature, it is evident that many works focussed on improving object detection but failed in detection of overlapped and occluded cases. In this context it is evident there is a gap to detect and classify the ripen and unripen tomatoes using a novel algorithm to overcome the issue, an improved tomato detection and classification algorithm (CAM-YOLO) is proposed. Further, the following section depicts the architecture and working of the proposed algorithm.

Methodology and Materials

Data collection

The images utilised in this study are collected from the Laboro Tomato dataset (LaboroAI, 2020), which is a tomato dataset consisting of tomatoes collected at various stages of their ripening developed for instance segmentation and object detection tasks. The dataset consists of 2,034 images with an input resolution of 416 × 416. It consists of images gathered under various environmental conditions such as occlusion, illumination, shading, overlap, and so on. Some of the images collected under various conditions are shown in Fig. 2.

Figure 2 (A) Single tomato. (B) Overlapping of tomatoes. (C) Tomatoes occlusion by branch. (D) Under shading conditions. (E) In sunlight conditions.

The YOLO object detection model requires labelled data, which includes the name and location of the ground truth bounding boxes in images. The labelImg is a graphical annotating tool is used for drawing the bounding boxes around every tomato in the image and labelling each ground truth bounding boxes with class label (ripe or unripe).

All visible tomatoes, whether ripe and unripe, were labelled with a bounding box in each image using the LWYS (Label What You See) approach. Notably, the bounding boxes for the highly occluded tomatoes were designed using the assumed shape based on the visible part of human intelligence.

Image augmentation

Before training the images, the dataset is augmented to artificially enhance its size. During training, each image is randomly sampled before being entered into the model using the following choices:

• flipping, rotating

• changing the brightness.

• original image

• thereby, the dataset is increased by 3 times.

Network architecture

Object detection techniques can be classified into two types: single-stage detectors and two-stage detectors. The first step in two-stage detector is to extract the candidate regions containing the target objects from the input image using selective search. The second step is to classify the extracted candidate region using regression analysis. The two-stage detectors typically include series of R-CNN models. The single stage detector, on the other hand, skips the candidate region extraction step and obtains object categorization (classification) and localization information in a single step. The single-stage object detectors are SSD, YOLO, DetectNet. Because single-stage object detectors outperform two-stage detectors in terms of inference, the article proposes an algorithm based on the YOLO object detection model.

The YOLO framework in shown in Fig. 3. The fundamental idea of the YOLO is to split the given image into S * S grids and detect the item (object) in each grid by predicting the bounding boxes and the confidence. Each grid needs to identify the bounding boxes along with their accuracy scores. If identified bounding box matches with the ground truth (GT) then the IoU (Intersection over Union) is equal to 1. This avoids bounding boxes that are not the same size as the true box.

Figure 3 YOLOv5 model.

Redmon & Farhadi (2017) introduced the YOLOv2 model in 2017 to automatically choose the best initial bounding box using K-means clustering method, thereby optimizing the object detection effect and speed in comparison to earlier versions. Later in 2018, they proposed another version YOLOv3 based on Darknet-53 for feature extraction. YOLOv3 motivated by FPN (Feature Pyramid Network) used to predicts the objects in three different scales. In 2020, Bochkovskiy, Wang & Liao (2020) proposed an updated version YOLOv4 to provide more substantial improvements in object detection. However, the lightweight model YOLOv5 proposed by Glenn Jocher superseded the earlier versions because of its fast real-time object detection. Based on the network depth and width variations, YOLOv5 classified into different versions: YOLOv5l, YOLOv5m, YOLOv5s, YOLOv5x. The YOLOv5 model as shown in Fig. 4 consists of three parts: backbone, neck, and head. The backbone extracts the features form inputted image at different granularities. Then, the neck performs aggregation on the features extracted by backbone and passes to the next layer for prediction. Finally, the head generates the bounding boxes and predicts the class labels.

Figure 4 YOLOv5 model architecture.

Input

The input of the YOLOv5 adopts mosaic data augmentation (Bochkovskiy, Wang & Liao, 2020) which stiches randomly four images together using clipping, scaling for better performance in small object detection. The representation of mosaic data augmentation is shown in Fig. 5. YOLOv5 adaptively produces different prediction bounding boxes near to the ground bounding box with the help of non-max suppression (NMS). Because the photos are not uniform in size, the adaptive zooming method is used to zoom them to an appropriate uniform size before being fed into the network for detection, avoiding concerns like a conflict between the feature map and the fully connected layer.

Figure 5 Mosaic data augmentation.

Backbone

The backbone of YOLOv5 uses the Focus layer for down-sampling to perform slice operation. The original RGB input image is fed into the Focus layer, and a slice operation is conducted to produce a 12-dimensional feature map, and then performs convolution operation using 32 kernels which results in 32-dimensional featue map.

YOLOv5 include two kinds of Cross Stage Partial structures (CSP): CSP1_X and CSP2_X. The first CSP struture named CSP1_X, is utilised in backbone to achieve rich gradient information by reducing the computation cost. Spatial Pyramid Pooling (SPP) is used in backbone to output feature maps of fixed size while maintaining image detection accuracy.

Neck

The YOLOv5 neck is primarly used to construct the feature pyramids, to improve the model’s detection of objects of various sizes, and realise recognition of the same object of various sizes and scales. YOLOv5 uses CSP2_X structure and Feature Pyramid Network (FPN) (Lin et al., 2017) and Path Aggregation Network (PAN) (Liu et al., 2018) as neck to aggregate the features.

Head

The YOLOv5 head comprises of nom-max supression and loss function. The loss function contains three parts: bounding-box loss, confidence loss and classification loss. The bounding box loss is calculated using the Generalized IoU (GIoU) (Rezatofighi et al., 2019). YOLOv5 leverages weighted NMS in the post-processing of target object detection, to filter multiple target bounding boxes and remove duplicate boxes.

Improved YOLOv5 model

This section discusses in detail about the proposed improved YOLOv5 architecture shown in Fig. 6 with attention mechanism and distance intersection over union. Because the CBAM attention module was developed to enhance key features, the proposed CAM-YOLO incorporated it into the network structure after each feature fusion, i.e., after “add,” “concat,” and before the detection head.

Figure 6 Proposed YOLOv5 architecture with attention mechanism.

Attention mechanism

The attention mechanism extracts a small portion of significant information from a large quantity of information and focuses on it, avoiding the majority of the unimportant information. The proposed algorithm CAM-YOLO makes use of convolutional block attention module (CBAM) (Zhu & Yang, 2018) which has two sub-modules: spatial and channel modules. Figure 7 depicts the CBAM structure.

Figure 7 CBAM structure.

The channel attention module attempts to capture “what” is significant in the given images, whereas the spatial attention module concentrates on “where,” or which section of an image is significant (spatial).

The channel attention module, as shown in Fig. 8, initially aggregates the spatial data of the given input feature map in 2 dimensions (height and width, respectively) by global max pooling and global average pooling. The output is then processed by the multilayer perceptron with the help of a shared fully connected layer (Desai & Shah, 2021). Finally, the final channel attention feature is built using the sigmoid activation operation and which is passed as input to the spatial attention module. The channel attention is computed as shown in Eq. (1). (1) MCAIF=σMLPMaxPoolIF+MLPAvgPoolIF.

The spatial attention module is introduced after the channel attention. In the channel dimension, the spatial attention module performs the max and mean pooling respectively (Fig. 9). First, we apply max-pooling and average pooling to the convolution module’s input. Second, as suggested by authors in Zhu & Yang (2018), the max-pooled and average-pooled tensors are concatenated. Then, convolution operation with kernel of size ( 7 × 7) and sigmoid activation function is used to activate the visual clues in images. The spatial attention map M SA(IF) is calculated using the Eq. (2)

Figure 8 Channel attention module.

Figure 9 Spatial attention module.

The proposed algorithm preserves the original network topology for the YOLOv5 backbone and extracts the features of the three feature layers of the backbone network and transmitting them to the head network. Next, the proposed algorithm incorporates CBAM in the head to reinforce the attention of the delivered feature map, so strengthening the network as it sends from the shallow layer to the deep layer. The learning of meaningful feature maps, particularly for small target features, can help the network learn the target’s feature information more effectively, capture the target’s recognition features more accurately in the same test image, and achieve a better recognition effect without increasing the training cost.

Improved non-max suppression

The objective of the non-max suppression in object detection algorithms is to choose the optimal bounding box for the object while suppressing all other boxes. Traditionally, the non-max supression calculates the IoU between the detection box with the best score and other boxes and deletes the boxes which have IoU larger than the specified threshold. By depending on the basic IoU, the NMS sometimes discards the occluded items incorrectly. Further, to enhance the missed detection scenario, Distance IoU(DIoU) (Zheng et al., 2020) is combined with NMS. When the two boxes have a significant IoU, it implies that two objects have been discovered, hence the boxes should not be deleted. DIoU considers the distance between the center point of the prediction and real bounding box, overlap rate making the regression more consistent, as shown in Eq. (2). (2) DIoU=IoU−ρ2b,bgtc2.

Where b, bgt denotes the central points of the predicted box(B) and ground-truth-box (Bgt), ρ2 denotes the Euclidean distance and IoU can be defined by using the Eq. (3). (3) IoU=B∩BgtB∪Bgt.

The DIoU definition can be formulated as shown in Eq. (4): (4) si=si,DIoUM,Bi<ɛ0,DIoUM,Bi≥ɛ.

Where si denotes the confidence score, M denotes the box with highest confidence score, Bi is all contrastive boxes of current class. In comparison to IoU, DIoU considers the information about the centre points of the two frames, which contributes to the resolution of the occluded issue produced by the close proximity objects. When evaluated by DIoU, the NMS effect is therefore more realistic and has a considerable influence.

Experiments and Results

Experimental setup

The experiment was conducted on Google Colab with GPU environment Tesla T4, and a total of 2,034 photos after augmentation of size 416 × 416 were used. The 70% of dataset is considered for training, remaining 30% for validation and testing purpose. The momentum and weight decay rates were set to 0.9 and 0.0005, respectively, and the experiment lasted 100 epochs. The IoU threshold is set to 0.5 during the testing phase.

Evaluation metrics

The effectiveness of the model is evaluated using precision, recall (Johnson et al., 2021) and mean average precision (mAP). Precision is determined by dividing the total number of positively identified samples by the proportion of accurately classified positive samples (True Positive) (either correctly or incorrectly). The recall is determined as the proportion of positive samples that were properly identified as positive to all positive samples. The recall assesses how well the model can identify positive samples. The precision and recall are calculated as follows: (5) Precisionp=TpTp+Fp

(6) RecallR=TpTp+Fn.

The average precision (AP) is defined as the region under the precision–recall curve, which denotes the average accuracy. (7) AP= ∫01PRdR

The mean average precision (mAP) is the aggregate of the AP for different categories and N is the number of categories and is formulated as: (8) mAP=∑1NAPN.

Results

The evaluation in this article is based on the loss function curve (loss) and average accuracy value (mAP) to accurately assess the effectiveness of the detection model.

In this section, the performance of the improved YOLOv5 with CBAM and DIoU is evaluated. The tomato dataset is divided into three parts: training, validation, and test sets in the ratio of 8:1:1 The experiment conducted on tomato dataset using YOLOv5s, YOLOv5-CBAM and DIoU to perform comparative studies. The loss function can directly depict whether the model can converge to stable point as the number of epochs increases as shown in Fig. 10.

Figure 10 The curve of training loss function.

The different performance metrics of the proposed improved YOLOv5 model for 100 epochs during training and validation are depicted in the Fig. 11.

Figure 11 Performance metrics of proposed model.

The proposed model reached its maximum in terms of precision, recall, and mean average precision after roughly 50 epochs. The objectness, box and classification losses were significantly decreased for first 50 epochs of model training and stabilised at around 75. As a result, we select the optimal weight for tomato detection and classification after training for 100 epochs.

In this study, the proposed CAM-YOLO algorithm is compared to common object identification approaches to objectively assess its benefits. To ensure fairness, all algorithms in the study makes use of same training parameters and samples, and the results are shown in Table 1.

Table 1 Comparison of detection algorithms.

Each row in the table shows the average precision, precision, recall of three models.

Algorithm	mAP@0.5%	Precision%	Recall%		
YOLOv5	85.9	84.9	80.1		
YOLOv5+ BottleneckCSP(backbone)	86.9	85.4	81.3		
CAM-YOLO	88.1	87.3	86.9		

From the Fig. 12 it is observed that the mAP @50% of the proposed algorithm CAM-YOLO is improved compared to YOLOv5 and YOLOv5 with BottleneckCSP as its backbone.

Figure 12 Comparison of detection algorithms.

Evaluation of detection results

To evaluate the model’s performance, numerous images from the dataset were chosen for testing, and the detection performance of the CAM-YOLO and traditional YOLOv5 models are shown in the following figures under various circumstances. Figure 13 depicts the identification of the overlapped tomatoes, where the occluded and overlapped tomatoes by the branch or other tomatoes is not identified in Fig. 13B, but the overlapped tomato is detected in Fig. 13C. Figure 14 depicts the identification of the small tomatoes, where the small tomato is not detected in Fig. 14B, but the same is detected in Fig. 14C using improved YOLOv5 model.

Figure 13 Identification of overlapped tomatoes (A) Original image (B) YOLOV5s (C) Improved YOLOv5 algorithm.

Figure 14 Identification of small tomatoes (A) Original image (B) YOLOV5s (C) Improved YOLOv5 algorithm.

To summarise, the CAM-YOLO model surpasses the traditional YOLOv5 when it comes to detecting tiny and dense targets, resulting in improved accuracy, detection, and identification.

The accuracy of the proposed algorithm may increase if more number of images were considered.

Conclusions

This article proposes a CAM-YOLO which is YOLOv5-based improved tomato detection and classification algorithm. First, started adding the attention module (CBAM) to the backbone allowed to extract the information quickly. Finally, the algorithm makes use of the DIoU with NMS to decrease the rate of missing the detection of overlapping tomatoes. The mAP@0.5 of the proposed algorithm is 88.1% which is an improvement compared to YOLOv5 model. Also, the proposed CAM-YOLO is efficient in addressing the low inference, accuracy and rate of missed target detection caused by overlapping and occlusion. In the future, CAM-YOLO algorithm can be utilised for the detection and classification of other fruits and vegetables, further different colored objects can also be categorised. For determining the metabolic changes that occur during ripening can be considered. To determine a tomato’s maturity, colour is one of the main criteria. However, this aspect differs from one cultivar to another even at the same maturity stage.

Additional Information and Declarations

Competing Interests

Author Contributions

Data Availability

The authors declare there are no competing interests.

Seetharam Nagesh Appe conceived and designed the experiments, performed the computation work, prepared figures and/or tables, authored or reviewed drafts of the article, and approved the final draft.

Arulselvi G. analyzed the data, authored or reviewed drafts of the article, and approved the final draft.

Balaji G.N. performed the experiments, authored or reviewed drafts of the article, and approved the final draft.

The following information was supplied regarding data availability:

The data is available at Github and Zenodo:

https://github.com/nageshappe/2048/blob/master/Yolov5_Tomato_detection.ipynb.

SeetharamNagesh Appe. (2023). Tomatoes dataset [Data set]. Zenodo. https://doi.org/10.5281/zenodo.7955736.

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
