# Peer review of "CAM-YOLO: tomato detection and classification based on improved YOLOv5 using combining attention mechanism"

_PeerJ Computer Science, doi:10.7717/peerj-cs.1463_

## Round 0.1 · original submission · Major Revisions

Dear Authors,

This is good work!

However, we have some suggestions to add quality to the article. Please update those corrections and re-submit it for the next review.

Reviewer 1 ·

Basic reporting

>> The language usage throughout this paper need to be improved, the author should do some proofreading on it.
>> Your abstract does not highlight the specifics of your research or findings. Rewrite the Abstract section to be more meaningful. I suggest to be Problem, Aim, Methods, Results, and Conclusion.
>> Introduction section can be extended to add the issues in the context of the existing work and how proposed algorithms/approach can be used to overcome this.
>> The problems of this work are not clearly stated. There is ambiguity in statement understanding.
>> Add main contributions list as points in the Introduction section.
>> Add the rest organization section at the end of the Introduction section.
>> More clarifications and highlighted about the research gabs in the related works section.
>> I feel that more explanation would be need on how the proposed method is performed.
>> What are the key challenges in detecting tomato fruits using classic deep learning object detection approaches like YOLO?
>> How does the proposed CAM-YOLO algorithm address these challenges?
>> Can you explain the role of Convolutional Block Attention Module (CBAM) in the CAM-YOLO algorithm?
>> How does Non-Maximum Suppression help in enhancing the identification of overlapping objects in the image?
>> What is DIoU (Distance Intersection Over Union) and how is it used in the CAM-YOLO algorithm?
>> What is the average precision achieved by the CAM-YOLO algorithm in detecting small and overlapped tomatoes?
>> Can the CAM-YOLO algorithm be used in other applications beyond tomato detection?
>> What is the size of the dataset used for training and validation of the CAM-YOLO algorithm?
>> How does the performance of CAM-YOLO compare to other state-of-the-art object detection algorithms?
>> Authors should add the parameters of the algorithms.
>> A comparison with state of art in the form of table should be added
>> Results need more explanations. Additional analysis is required at each experiment to show the its main purpose.
>> The Limitations of the proposed study need to be discussed before conclusion.
>> Rewrite the Conclusion section to be:
- You must more clearly highlight the theoretical and practical implications of your research
-Discuss research contributions.
-Indicate practical advantages (in at least one separate paragraph),
-discuss research limitations (at least one separate paragraph), and
-supply 2-3 solid and insightful future research suggestions.

Experimental design

My comments are provided in the basic report

Validity of the findings

My comments are provided in the basic report

Reviewer 2 ·

Basic reporting

Needs improvement in figure resolutions, equation formatting, and structure of the headings.

Experimental design

The experiment is an improved version of an existing model. The work seems exciting for detecting tomato classes but the authors are encouraged to do check the existing classification methods in future work.

Validity of the findings

Findings are satisfactory.

·

Basic reporting

In this work, the authors evaluate the YOLOv5 models for tomato detection and classification with improved attention. This work has shown some significant contributions and could be helpful for researchers working in the field of object recognition and classification tasks. However, there are some issues in this report which are listed as follows:
1. It is recommended to make some taxonomy figures in the introduction section for assisting readers to understand your work visually.
2. Your literature section (related work) needs more detail. I suggest you include the justification regarding the selection of a two-stage detector model in your work instead of a single-stage detector model.
3. Zhu et al (https://doi.org/10.1007/s11119-023-09992-w) have proposed similar architecture. Authors need to justify the nobility in their proposed work.
4. There are multiple typo errors and missing spaces (e.g. lines 35, 86, 230, and more).
5. Equation 4 contains a “?” sign, please justify this symbol.

Experimental design

no comment

Validity of the findings

no comment

Additional comments

no comment

---

## Round 0.2 · accepted · Accept

Dear Author/Authors,

Good Work and keep it up

Reviewer 1 ·

Basic reporting

The authors did all the corrections required and the manuscript can be accept in this form

Experimental design

The authors did all the corrections required and the manuscript can be accept in this form

Validity of the findings

The authors did all the corrections required and the manuscript can be accept in this form

Reviewer 2 ·

Basic reporting

No comments.

Experimental design

No comments.

Validity of the findings

No comments.

Additional comments

No comments.

·

Basic reporting

My concerns were addressed by the authors, and they incorporated my suggestions into the revised manuscript.

Experimental design

none

Validity of the findings

none

Additional comments

none